# Genetic Diversity and Antimicrobial Resistance of Extraintestinal *E. coli* Populations Pre- and Post-Antimicrobial Therapy on Broilers Affected by Colisepticemia

**DOI:** 10.3390/ani13162590

**Published:** 2023-08-11

**Authors:** Frédérique Pasquali, Cecilia Crippa, Antonio Parisi, Alex Lucchi, Lucia Gambi, Alessandra Merlotti, Daniel Remondini, Maurizio Stonfer, Gerardo Manfreda

**Affiliations:** 1Department of Agricultural and Food Sciences, Alma Mater Studiorum–University of Bologna, 40126 Bologna, Italyalex.lucchi3@unibo.it (A.L.); luciagambi@gmail.com (L.G.); gerardo.manfreda@unibo.it (G.M.); 2Istituto Zooprofilattico Sperimentale di Puglia e Basilicata, 71121 Foggia, Italy; antonio.parisi@izspb.it; 3Department of Physics and Astronomy, Alma Mater Studiorum–University of Bologna, 40126 Bologna, Italy; alessandra.merlotti2@unibo.it (A.M.); daniel.remondini@unibo.it (D.R.); 4Bayer Animal Health, 20156 Milan, Italy; maupost.it@gmail.com

**Keywords:** colibacillosis, extraintestinal *E. coli*, enrofloxacin, resistome, plasmid persistence

## Abstract

**Simple Summary:**

Although the use of antimicrobials and the related selection of antimicrobial-resistant (AMR) pathogens are recognized worldwide, limited or no information is available on the effect of the antibiotic treatment on the genetic structure as well as the dissemination and persistence of plasmids carrying AMR genetic determinants. This is of particular interest for bacterial pathogens like Avian Pathogenic *Escherica coli* (APEC), which have the potential to persist and be transmitted from broilers to humans through the food chain as well as for their AMR genetic determinants, which have the potential to be mobilized and spread. With a genomic approach, results of the present study revealed that during the enrofloxacin treatment of broilers, initial strains of extraintestinal *E. coli* disappeared, being substituted by new clones. Plasmid-mediated fluoroquinolone resistance did not appear to disseminate or persist among observed genomes, confirming that QRDR mutations rather than plasmids are the main drivers of quinolone resistance. Interestingly, plasmids carrying other AMR genes than quinolone-determinant ones were disseminated and persistent since they were found in different clones both before and after the treatment. The persistence of plasmids without a direct antimicrobial selective pressure, if confirmed with further studies, might give insights on the so-called plasmid paradox.

**Abstract:**

The aim of the present study was to investigate the genetic diversity and antimicrobial resistance (AMR) of *E. coli* during enrofloxacin therapy in broilers affected by colisepticemia. Three unrelated farms with ongoing colibacillosis outbreaks were sampled at day 1 before treatment and at days 5, 10 and 24 post-treatment. A total of 179 *E. coli* isolates were collected from extraintestinal organs and submitted to serotyping, PFGE and the minimum inhibitory concentration (MIC) against enrofloxacin. PFGE clusters shifted from 3–6 at D1 to 10–16 at D5, D10 and D24, suggesting an increased population diversity after the treatment. The majority of strains belonged to NT or O78 and to ST117 or ST23. PFGE results were confirmed with SNP calling: no persistent isolates were identified. An increase in resistance to fluoroquinolones in *E. coli* isolates was observed along the treatment. Resistome analyses revealed *qnrB19* and *qnrS1* genes along with mutations in the *gyrA*, *parC* and *parE* genes. Interestingly, despite a fluoroquinolone selective pressure, *qnr*-carrying plasmids did not persist. On the contrary, two conjugative AMR plasmid clusters (AB233 and AA474) harboring AMR genes other than *qnr* were persistent since they were identified in both D1 and D10 genomes in two farms. Further studies should be performed in order to confirm plasmid persistence not associated (in vivo) to antimicrobial selective pressure.

## 1. Introduction

*Escherichia coli* is a commensal bacterium of the gastrointestinal tract of humans and animals. However, in some circumstances, *E. coli* can cause extraintestinal diseases. In broilers, colibacillosis is localized or systemic and it is associated with post-mortem lesions such as airsacculitis, pericarditis, perihepatitis and peritonitis [1]. This disease causes significant economic losses for world poultry producers [2]. Mortality rates ranging from 3.3% up to 28.3% have been registered worldwide [3,4,5,6,7,8]. In addition, an even higher concern is associated with the similarities observed between Avian Pathogenic *Escherichia coli* (APEC) and human pathogenic *Escherichia coli*, thus suggesting a zoonotic potential [9,10,11].

*E. coli* shows a high intra-population diversity [12,13,14,15]. In terms of the serogroup, a great variety has been described worldwide, although the primary serogroups associated with APEC include O1, O2 and O78 [16]. Serogroups O19, O35, O84, O142 and O157 were also detected among APEC isolates, although to a lesser extent [17,18,19,20]. A high rate of untypable strains was also observed [17,20]. In terms of Multi-Locus Sequence Typing (MLST), the most frequently reported sequence types (ST) were ST117, ST23 and ST350. Other ST were ST48, ST95, ST140, ST141, ST155, ST162, ST428, ST949 and ST1618 [12,16,17,21,22].

Unfortunately, the use of antimicrobials over the last few decades has resulted in the selection of multi-drug-resistant strains dramatically reducing treatment options [21,23]. In the last decade, in Italy, among 110 APEC isolated from broilers with colibacillosis, high levels of resistance were observed for sulphonamides (76.4%), streptomycin (62.7%) and sulphamethoxazole combined with trimethoprim (63.6%) [24]. In the same country, high rates of occurrence of resistant APEC against oxytetracycline were also described [25]. Worldwide, percentages of resistance ranging between 63.5% and 91% were reported for APEC, isolated from chickens, against ampicillin, amoxicillin, streptomycin, chloramphenicol, tetracycline, nalidixic acid and ciprofloxacin [21,26].

Although the use of antimicrobials and their effects on the selection of antimicrobial-resistant pathogens are recognized worldwide, limited or no information is available on the effect of an antibiotic treatment on the genetic structure and antimicrobial resistance patterns of a bacterial population. This is of particular interest for bacterial pathogens like APEC, which have the potential to persist and be transmitted to humans as well as for their AMR genetic determinants, which have the potential to be mobilized and spread among other Gram-negative communities.

The aim of the present study was to investigate the genetic diversity of extraintestinal *E. coli* populations during antimicrobial therapy in broilers affected by colisepticemia. Additionally, a selection of newly sequenced *E. coli* genomes of isolates collected before, during and after the treatment were compared in order to gain insights on their genomic relatedness, as well as to compare their resistome and its potential mobilization.

## 2. Materials and Methods

### 2.1. Sample Collection

Three unrelated conventional broiler farms were selected. Two farms (Farms A and C) with a daily mortality rate over 0.2%, and one farm (Farm B) with a daily mortality rate of 0.12%, were identified in Northern Italy in 2016. In all three commercial farms, broilers showed symptoms of lethargy, a lower body weight increase and gross pathological lesions related to colisepticemia. In particular, Farm B was chosen, although showing a lower percentage of the daily mortality rate, because of the evidence of clinical signs of septicaemic colibacillosis. All animals were vaccinated at the hatchery at regular doses for Newcastle disease, infectious bronchitis, Marek disease and Gumboro disease and they were fed at libitum with feed containing coccidiostats. Broilers were submitted to antibiotic treatment with enrofloxacin in drinking water at a dose of 10 mg/kg of body weight. A total of ten animals were humanely euthanized by farm veterinarians from each farm on day 1 (D1) (start of treatment and prior to first dosing), D5 (end of treatment), D10 and D24 (day 10 and 24 after treatment; D24 only in Farm B). At D1, birds were 20, 14 and 37 days old in Farm A, B and C, respectively. The production cycle lasted 37 days in Farms A and B and 45 days in Farm C. After euthanasia with cervical dislocation, gross pathological lesions of carcasses were scored and samples of the lungs, central nervous system (cns), heart (pericardium layers) and spleen were collected for *E. coli* isolation. All samples were collected from farm veterinarians for diagnostic purposes.

### 2.2. Escherichia coli Isolation

Organ samples were streaked on MacConkey agar (Thermo Scientific, Milan, Italy) for the semi-quantitative evaluation of the presence of *E. coli* and incubated at 37 ± 1 °C for 24 h [27]. Single colonies showing characteristic *E. coli* morphology were selected from each plate, inoculated in a Brain Heart Infusion Broth (BHI, Thermo Scientific) and incubated at 37 ± 1 °C for 24 h. Pure cultures were stocked and stored at −80 °C with 10% of glycerol until further use. Following, each stock culture was submitted to biochemical tests with the API system (BioMérieux, Marcy L’Etoile, Lyon, France) for species confirmation. A total of 179 *E. coli* isolates were selected, all belonging to different sampled organs and distributed as follows: 52 from Farm A, 70 from Farm B and 57 from Farm C. The assigned ID was built on two numbers: the first number representing the day of sampling at one of the three farms and the second related to the animal.

### 2.3. Serotyping

All 179 *E. coli* isolates underwent conventional serotyping for O antigens according to standard procedures as previously described [28]. Commercial antisera of all 188 *E. coli* O antigens were used (SSI Diagnostica^®^, Milan, Italy). To remove nonspecific agglutinins, the antisera were adsorbed with the corresponding cross-reacting antigens.

### 2.4. Pulse-Field Gel Electrophoresis

A total of 179 *E. coli* isolates underwent molecular typing with pulsed-field gel electrophoresis (PFGE). The PFGE was performed following the PulseNet standardized protocol, using *Salmonella enterica* serovar Braenderup H9812 as a molecular size marker and *Xba*I as a restriction enzyme [29]. In particular, an inactivation step against DNAse was added to the protocol as previously described [30]. After importing into Bionumerics 7.1 software (Applied Maths, Saint-Martens-Latem, Belgium), profiles were normalized and compared using the Dice similarity index, and the unweighted-pair group method using the average linkage algorithm (UPGMA) was used to create the dendrogram, with a 1.0% optimization setting and 1.2% band position tolerance. Isolates showing a 95% PFGE similarity cut-off were considered as closely related and grouped in the same PFGE cluster.

### 2.5. Antimicrobial Susceptibility Testing

Antimicrobial susceptibility was tested with the microbroth dilution method as reported by the Clinical and Laboratory standard Institute [31]. *E. coli* ATCC 25,922 was included as a quality control. Isolates were tested against enrofloxacin (ENR). Twelve serial dilutions were tested from 0.016 to 32 mg/L and the following clinical break points were used: susceptible S ≤ 0.25 mg/L, intermediate (I) if included in the range 0.5–1 mg/L and resistant R ≥ 2 mg/L.

### 2.6. Whole-Genome Sequencing and De Novo Assembly

Within the 179 *E. coli* isolates, a representative dataset of 31 isolates was selected and whole-genome sequenced. The 31 isolates were selected based on different metadata (the farm, day of sampling and anatomic portion) as well as PFGE profiles and MIC values, in order to gather a comprehensive population diversity. Genomic DNA was extracted using the MagAttract HMW DNA Kit (Qiagen, Hilden, Germany). The DNA concentration and quality parameter ratio 260/280 were measured with BioSpectrometer fluorescence (Eppendorf). Libraries were built using a Nextera^®^ XT DNA Library Preparation Kit (Illumina, Milan, Italy) and sequenced on an Illumina MiSeq Platform, which generates tagged 250 bp paired-end reads. Reads were de novo-assembled, applying INNUca v3.2, a fully automated open-source pipeline where reads are quality checked with FASTQC, and read cleaning is performed with TRIMMOMATIC and de novo draft genome assembly is performed with SPAdes v3.9.0 [32]. Finally, the whole draft genome was improved using PILON by correcting bases and fixing misassemblies. Moreover, INNUca includes the MLST tool, which assigns the 7-loci MLST Sequence Type based on PubMLST typing schemes. Read sequences are available at the European Nucleotide Archive under the study accession number PRJEB36793.

### 2.7. In Silico Analyses

De novo assemblies of the 31 selected genomes were submitted to the ABRicate tool, which performs a BLAST search of genes included in specific databases [33]. The Resfinder Database was selected for the genomic characterization of the resistome. Chromosomal point mutations associated with antimicrobial resistance were investigated using ResFinder v.4.0 [34]. Abricate results were sorted and genes with an identity equal to or higher than 90% and coverage equal to or higher than 60% were retained and considered positive for the gene. FimH typing was performed using FimTyper v.1.0 [35,36]. The localization of antimicrobial resistance-determinant genes on conjugative, mobilizable or non-mobilizable plasmids or on the chromosome was investigated by using MOB-Recon tool v.3.0.1, which provided additional typing information about plasmids [37].

### 2.8. SNP Calling

Single nucleotide polymorphism (SNP) calling was performed with Snippy, a rapid haploid variant calling and core genome alignment open-source tool [38]. The pipeline includes several tools that align reads or assemblies from each isolate to a reference genome and then identifies variants among the alignments. Based on the core SNP alignment of the 31 *E. coli* isolates sequenced in the present study, a high-resolution phylogeny tree was built including the conserved nucleotide variant sites shared by all genomes. PhyML v3.3.2 was used to analyse the SNP differences between isolates based on the maximum likelihood algorithm and phylogenetic trees were visualized with iTOL [39]. Finally, a pairwise SNP distance matrix was built using snp-dists v0.6.3 [40].

### 2.9. Data Analysis

Data were analysed using Excel (Microsoft, 2016 version). A one-way ANOVA test was used to compare data of the three groups (D1, D5 and D10).

## 3. Results

### 3.1. E. coli Isolation

One-hundred and seventy-nine *E. coli* were collected as single isolates from each of the following organs: the lungs, spleen, heart and cns (Table 1). Interestingly, in the spleen, heart and cns, the number of isolated *E. coli* decreased significantly from D1 to D5 (*p* = 0.00699), which is in line with the reduction in the scores of gross pathological lesions of carcasses.

### 3.2. Serotyping

Within the 179 *E. coli* isolates, the large majority was not typable (NT) (127) (Table 2). Among typable isolates, the serogroup most frequently identified was O78 followed by O86 (5) and O157 (3). Regarding the 40 O78 isolates, all isolated from Farm B, 36 were isolated at day 1 before antimicrobial treatment, suggesting this serogroup was the dominant strain likely infecting the flock. In the literature, the NT serogroup has already been observed as the most frequently identified APEC serogroup followed by O78 [16,17,20]. Interestingly, three isolates, collected from Farm C and isolated from the lungs, belong to serogroup O157, which includes enterohemorrhagic strains of a human health concern [41].

### 3.3. Pulsed-Field Gel Electrophoresis

According to PFGE, the genetic diversity was relatively low before treatment and increased thereafter (Figure 1, Figure 2 and Figure 3). Independently from the farm, at D1, *E. coli* isolates clustered in 3–6 clusters at a 95% similarity, whereas at D5, D10 and D24, the number of clusters increased to 10–16 with most clusters containing single isolates. Of note, more than one clone was collected at D1 in the same farm, suggesting that colibacillosis was associated with different APEC strains. Some PFGE profiles were found both in D1 and D5 isolates or D5 and D10 isolates. However, no PFGE profiles were recurrently found at all time periods, suggesting the lack of the persistence of D1 strains.

Based on the percentage of similarity of 95%, in Farm A, *E. coli* isolates collected before the antibiotic treatment (D1) were grouped in 6 PFGE clusters, whereas after the antibiotic treatment (D5 and D10), 14 PFGE clusters were detected (Figure 1). One D1 isolate (871-42) showed a similar PFGE pattern to a D5 isolate (916-43) at 95,7% of similarity (both collected from the lungs), whereas no PFGE patterns were shared between D1 and D10 isolates. One D5 isolate (916-45) showed a similar PFGE pattern to a D10 isolate (976-25) with 96% of similarity. Within isolates collected after the treatment (D5 and D10), a wide diversity of PFGE profiles was identified without any correlation to the date of collection or the organ of origin (Figure 1).

In Farm B, *E. coli* isolates collected at D1 clustered in three PFGE clusters. In particular, two of them included isolates from different organs (the heart, spleen and lungs) (Figure 2), suggesting that the same strain colonised different organs. Isolates collected after the treatment clustered in a higher number of PFGE clusters: 10, 7 and 2 patterns for D5, D10 and D24, respectively. No similar PFGE patterns were observed between isolates collected before and after treatment (Figure 2). Although, after the treatment, four D5 isolates (1665-22, 1665-42, 1665-44 and 1665-48) showed a similar PFGE pattern to D10 isolates (1684-33, 1684-49 and 1684-44) with similarities ranging from 96.8 to 100%.

In Farm C, *E. coli* isolates belonging to D1 clustered in 4 different patterns, whereas 11 different patterns were identified for D5 and D10. Three D1 isolates (2750-29, 2750-43 and 2750-59) shared the same PFGE profile as two D5 isolates (2835-38 and 2835-39) and showed a similarity of 98.9% with two additional D5 isolates (2835-28 and 2835-32). Moreover, one D5 isolate (2835-44) showed the same PFGE profile as five D10 isolates (2863-44, 2863-47, 2863-48, 2863-49 and 2863-50) (Figure 3).

In all three farms, no PFGE profiles persisted along the time of sampling (D1-24), suggesting that initial pathogenic D1 isolates disappeared, being effectively eradicated by the antimicrobial treatment and substituted by new isolates.

### 3.4. Minimal Inhibitory Concentration (MIC)

MIC values of 179 *E. coli* isolates against enrofloxacin are reported in Table 3. Not surprisingly, before the antibiotic treatment at D1, the majority of isolates were susceptible to enrofloxacin (67/91, 73.6%), whereas after the treatment at D5, D10 and D24, only four isolates showed MIC values lower than the breakpoint of susceptibility (4/88, 4.5%), suggesting a shift, although not significant (*p* = 0.1535), of the extraintestinal *E. coli* population from susceptibility to resistance induced by the selective pressure of the antimicrobial therapy. In association with PFGE findings, these results suggest the disappearance of an enrofloxacin-susceptible D1 population, which was replaced by new enrofloxacin-resistant *E. coli* clones.

### 3.5. Whole-Genome Sequencing

Based on metadata as well as MIC and PFGE data, 31 representative isolates were subjected to whole-genome sequencing and a de novo assembly. DNA extracted from isolates had a concentration ranging from 9.1 to 148.6 µg/mL and a 260/280 ratio from 1.80 to 1.99. The generated draft whole-genome sizes ranged from 4,849,691 to 5,533,085 bp. Except for one genome that showed outlier values, all draft genomes included from 99 to 770 contigs were with a coverage between 54 and 136X and N50 values ranging from 23,145 to 174,567 (Appendix A).

Regarding MLST, two ST types were frequently identified at D1 before the antibiotic treatment: ST23 and ST117 (Appendix A). Both ST types were previously described regarding UPEC and ETEC strains [12,16,22]. Other ST types observed at D1 were ST10, ST101, ST140, ST295, ST1618 and ST7080. According to FimH typing, a unique H type was identified for these ST types, namely ST117–H97 and ST23-H35 (Appendix A). Interestingly, one strain, isolated at D1 in Farm C, belonged to ST10—already described in Denmark as associated with a multiple-disease outbreak in the same broiler production over 18 months [42]. Regarding strains isolated after the antimicrobial treatment, two strains isolated at D10 in Farms A (976-38) and B (1684-52) belonged to ST43—already described as gathering extended spectrum β-lactamase (ESBL) strains [43].

#### 3.5.1. SNP Calling

The genetic relationship between the 31 *E. coli* isolates sequenced in the present study is reported in Figure 4.

*E. coli* isolates collected before the antibiotic treatment at D1 shared from 82 to 98,137 SNPs in Farm A, from 44 to 86 SNPs in Farm B and from 109 to 89,968 SNPs in Farm C (Appendix A). SNP data confirmed the hypothesis that multiple strains, and not only one, were the causative agents of colisepticemia in all three farms. In particular, in Farm B, SNP calling confirmed the presence at D1 of genetically related strains (all belonging to O78-ST23-H35), which probably share a common ancestor. As was the case for PFGE, also with SNPs, the higher genetic diversity among strains collected from D5 to D10 in comparison to D1 is confirmed (Figure 4 and Appendix A).

#### 3.5.2. Characterization of Genetic Determinants of Enrofloxacin Resistance

All enrofloxacin-resistant and intermediate-sequenced genomes carried two or more point mutations in the *gyrA* gene (S83L and D87N), the *parC* gene (S80R, S80I and E84G) or the *parE* gene (S458A). One genome collected at D5 in Farm B (1665-24) and two genomes collected at D1 (2750-48) and D5 (2835-47) in Farm C carried additional plasmid-mediated quinolone resistance (PMQR)-determinant genes *qnrB19* and *qnrS1* (Table 4). Both the identification of genes in associations to detected mutations as well as the identification of one or more detected mutations have already been described as associated with fluoroquinolone resistance [44].

#### 3.5.3. In Silico Resistome Characterization

Regarding antimicrobial resistance determinants other than fluorquinolone ones, sequenced genomes carried genes predicting multiresistance phenotypes (resistance to more than three antimicrobial agents) (Figure 5). In particular, except for the four D1 genomes of isolates collected at D1 in Farm B (ID 1614) predicted as resistant to aminoglycosides (*acc(3)-IId*) and β-lactams (*bla*TEM-1B), all genomes tested showed the co-occurrence of different antimicrobial-resistant-determinant genes predicting resistance against aminoglycosides (*aac(3)-IId*, *aadA*, *ant(3”)-Ia*, *aph(3”)-Ib* and *aph(6)-Id*), β-lactams (*bla*TEM), chloramphenicol (*catA* and *clmA*), trimethoprim (*dfrA*), lincosamides (*lnu*), sulphonamides (*sul1*, *sul2* and *sul3*) and/or tetracycline (*tet(34)*, *tet(A)* and *tet(B)*). Interestingly, multiresistant predicting genotypes were found both before and after the enrofloxacin treatment. This indication suggests that antimicrobial resistance genes circulated all along the period under study, independently from the enrofloxacin treatment and leading to the hypothesis of a potential dissemination of those determinants from the D1 population to the D5, D10 and D24 *E. coli* populations through the horizontal transfer of plasmids carrying those genes.

#### 3.5.4. Localization of Antimicrobial Resistance Genes

The majority of antimicrobial-resistance-associated genes were located on plasmids (Table 5). Most of the plasmid-mediated genes were conjugative or mobilizable (Table 5).

Regarding plasmid-mediated quinolone resistance (PMQR), *qnrB19* was located in a mobilizable plasmid and *qnrS1* in two conjugative plasmids (Table 5).

The *qnrB19*-carrying contig of 2744 bp showed a 98% coverage and 100% identity with the plasmid pRIVM_C014947_7 found in one strain of *Klebsiella pneumoniae* (accession number MT560070.1) and plasmid pUWI-PP122.1 found in one strain of *Salmonella enterica* (accession number CP066326.1) (Figure 6).

The *qnrS1*-carrying contigs of isolate 2750-48 and isolate 2835-47 showed a 100% coverage and 99,98% identity with plasmid sequences of *Klebsiella flexneri* (accession numbers CP012734.1 and CP020341.1), *E. coli* (accession number MH121702.1) and *Salmonella enterica* serovar Dessau (accession number CP043765.1). The quinolone-resistance-carrying contigs (*qnrS1* and qnrB19) are displayed in Figure 7; notably, the *qnrS1* gene harboured by the 2835-47 strain is located in one integron closely located to a Tn*3* transposon carrying the *bla*TEM gene (Figure 7).

Interestingly, plasmid clusters AB233 in Farm A and AA474 in Farms B and C showed persistence, being identified both before (D1) and after (D5 and/or D10) the antimicrobial treatment (Table 5). Plasmid cluster AB233 harboured *bla*TEM-1B, *catA* and *tet(A)* genes. The AA474 plasmid cluster carried *bla*TEM-1B, *aac(3)-IId*, *sul2* and *dfrA1* associated, respectively, with β-lactam, aminoglycoside and trimethoprim/sulphonamide resistance. None of the two plasmid clusters harboured fluroquinolone-resistance-determinant genes.

## 4. Discussion

In the present study, the genetic diversity and antimicrobial resistance of *E. coli* isolated from the spleen, pericardium, lungs and cns during enrofloxacin therapy on broilers affected by colicepticemia were evaluated. In three conventional farms, a clear temporal variation of the *E. coli* population was observed. Along the treatment, the number of *E. coli* isolates collected from extraintestinal organs decreased and the *E. coli* diversity population increased with a disappearance of the predominant genotypes identified before the treatment and the emergence of new diverse genotypes. This higher genetic diversity might be associated with the disappearance of the pathogenic clone and the emergence of different apathogenic *E. coli* ones. The disappearance of pathological lesions on selected organs after the treatment reinforces this hypothesis.

Whilst serotyping still represents the most applied diagnostic method for APEC identification, only 29% of the isolates positively reacted with commercial antisera, confirming 078 as the most reported serotype (77% of typable strains). The high rate of untypable isolates observed in this study is consistent with other reports [17,20], addressing various limitations to the implementation of serotyping as an effective diagnostic tool. However, the whole-genome sequencing approach carried out on a representative subset of 31 strains allowed for the identification and subtyping of APEC strains with a greater accuracy than the conventional serotyping methods. Within D1 *E. coli* isolates, two ST types were the most represented: ST117 and ST23. ST117 has been observed in broilers and breeders in Nordic poultry production [22]. APEC and UPEC isolates, belonging to ST117, have been described, suggesting this ST as potentially zoonotic [12]. Similarly, based on their genome content, two ST23 APEC isolates were more closely related to the genomes of human enterotoxigenic ST23 *E. coli* than to APEC O1-ST95 [16].

Regarding antimicrobial resistance, in all farms, high percentages of *E. coli* strains were resistant to beta-lactams, fluoroquinolones, tetracycline and trimethoprim/sulphamethoxazole. Those resistances are in line with the use of these antimicrobials in poultry in the whole EU [23]. Further studies on antibiotic-free farms will be interesting to compare the occurrence of antimicrobial resistance phenotypes and genotypes.

An increasing trend of enrofloxacin-resistant isolates was detected from D1 to D10. This observation suggests that enrofloxacin therapy selected fluoroquinolone-resistant *E. coli* isolates (cut-offs R ≥ 2 (mg/L)). Further studies on a longer period of time post-treatment are required to investigate whether those selected resistant phenotypes persist over time in the absence of a selective pressure.

According to WGS results, point mutations were detected in all enrofloxacin-resistant sequenced isolates. In particular, mutations were found in the *gyrA* gene, the *parC* gene and the *parE* gene. The combination of these mutations has been known to be associated with fluoroquinolone resistance for many years [45]. Along with these mutations, three isolates carried *qnrB19* or *qnrS1* genes. The *qnr* genes were previously detected in *S. enterica*, *E. coli*, *Klebsiella pneumoniae* and other clinical and environmental isolates [46,47,48]. Although rarely found in the past, in more recent years, they are re-emerging as a concern due to their increased frequency of detection in some countries [48]. Moreover, although associated with a phenotype of a reduced susceptibility in the absence of additional QRDR mutations, they have been recently associated with clinical resistance to ciprofloxacin under urinary tract physiological conditions [49].

The majority of antimicrobial-resistance-associated genes detected in the present study were localized in conjugative or mobilizable plasmids, suggesting their potential spread to other bacteria with horizontal gene transfer. Interestingly, two plasmid clusters, carrying beta-lactam, tetracycline and trimethoprim/sulphametoxazole resistance genes but not fluoroquinolone resistance ones, were found as repeatedly isolated before and after the antimicrobial treatment, suggesting the persistence and spread of these plasmids. The persistence of plasmids without a corresponding antimicrobial selection pressure has been addressed as the plasmid paradox [50]. If, on one side, without a selective pressure, costly plasmids should be lost over time, the current study confirms (in vivo) that plasmids can persist, even in the absence of positive selection, as was already observed in vitro [51,52,53]. Further studies will be needed to clearly evaluate the persistence of those plasmids in a longer time period as well as to decipher the molecular bases of these observations.

## 5. Conclusions

In conclusion, in the present work, the population of extraintestinal *E. coli* during and after the antimicrobial treatment in broilers affected by colisepticemia revealed an increasing genetic diversity. Interestingly, phylogenetic and genomic analyses did not reveal the persistence of one strain all along the study period, suggesting that the extraintestinal enrofloxacin-susceptible *E. coli* population found before the treatment left room for the colonization of a new and more diverse enrofloxacin-resistant *E. coli* population. If isolates did not persist, AMR plasmid did. Genomic analyses revealed the persistence of two conjugative plasmid clusters, suggesting that the enrofloxacin treatment was effective in eradicating the initial extraintestinal *E. coli* population; however, plasmids carried by this population were able to persist and spread to the extraintestinal population found after the treatment.

## Figures and Tables

**Figure 1 animals-13-02590-f001:**
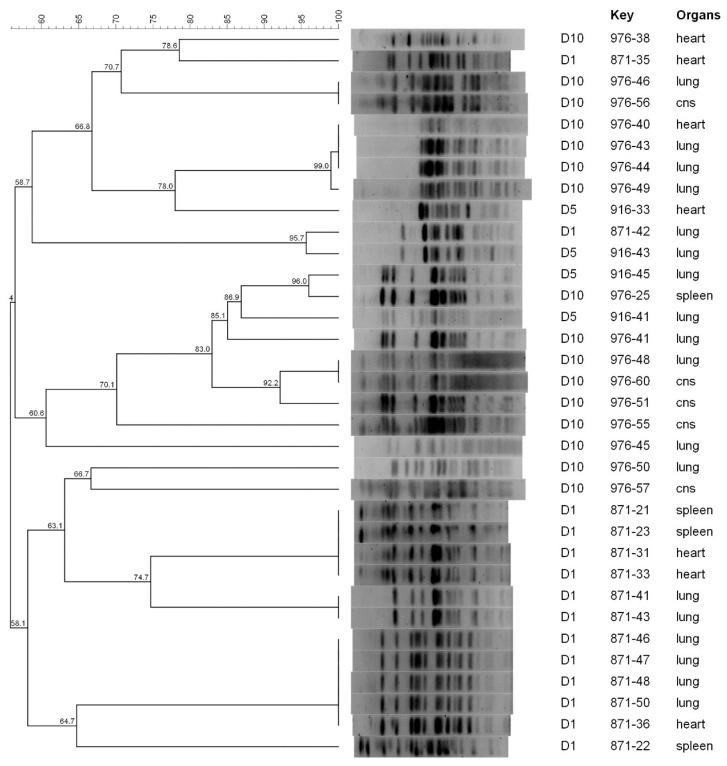
PFGE dendrogram of *E. coli* isolates collected from broilers in Farm A.

**Figure 2 animals-13-02590-f002:**
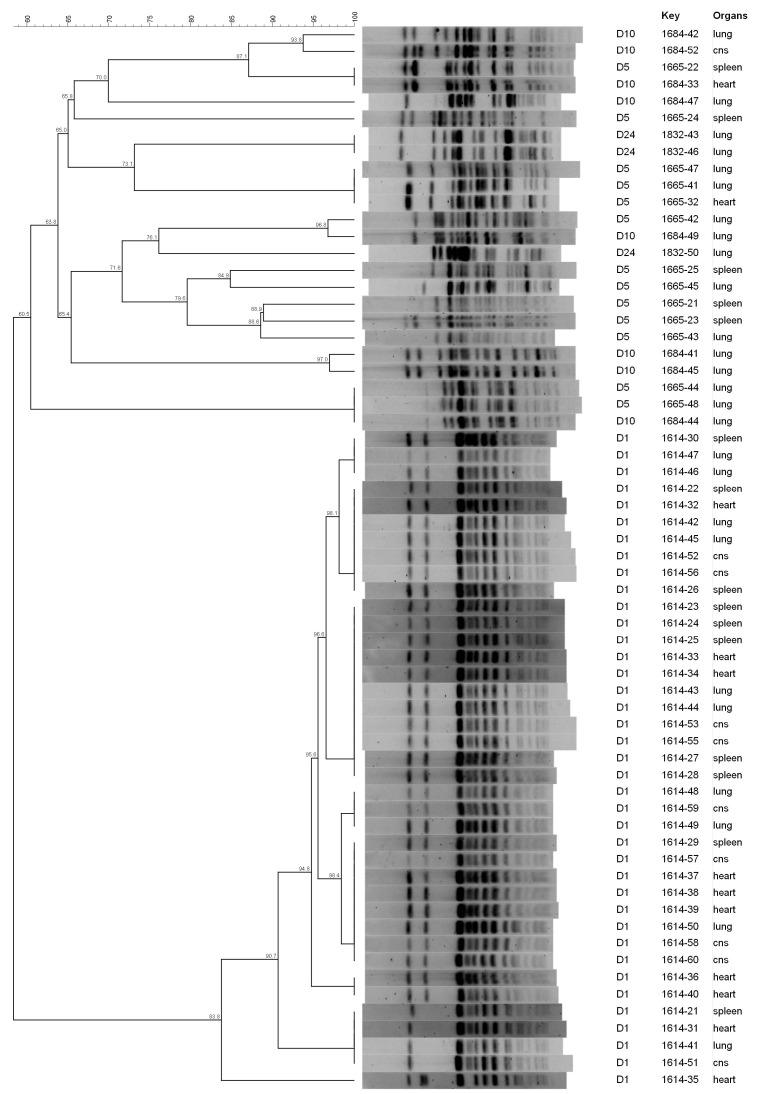
PFGE dendrogram of *E. coli* isolates collected from broilers in Farm B.

**Figure 3 animals-13-02590-f003:**
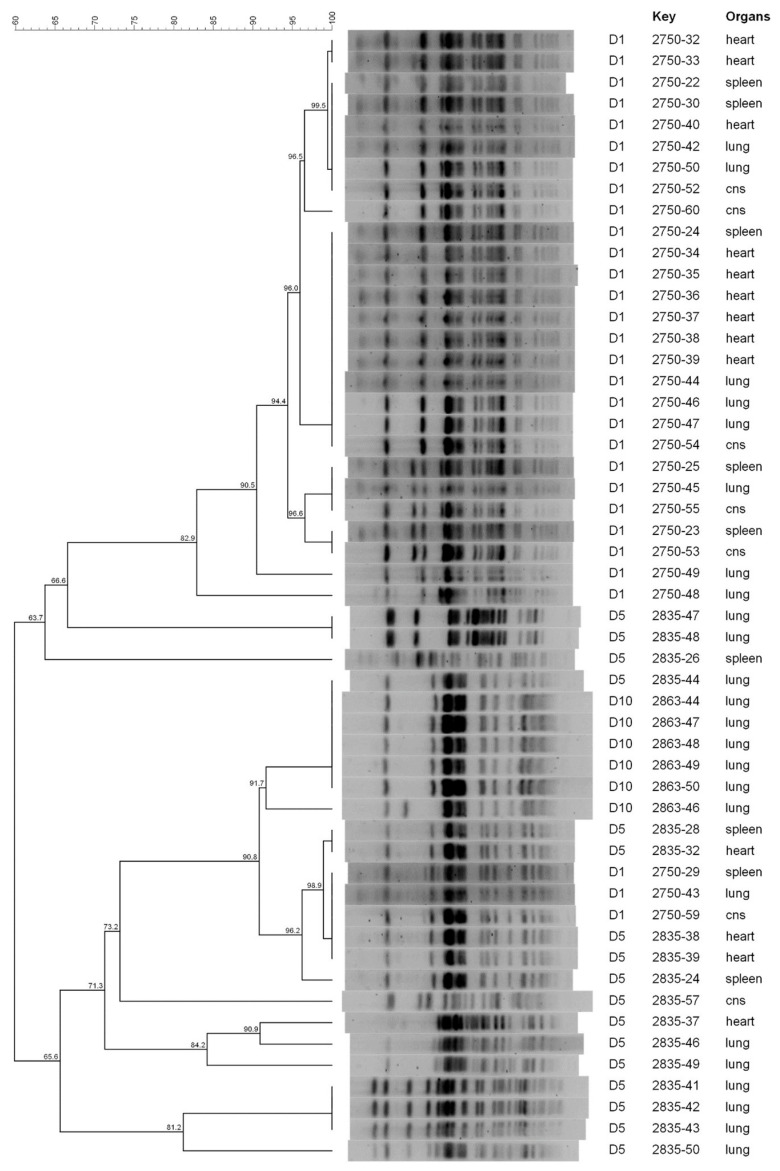
PFGE dendrogram of *E. coli* isolates collected from broilers in Farm C.

**Figure 4 animals-13-02590-f004:**
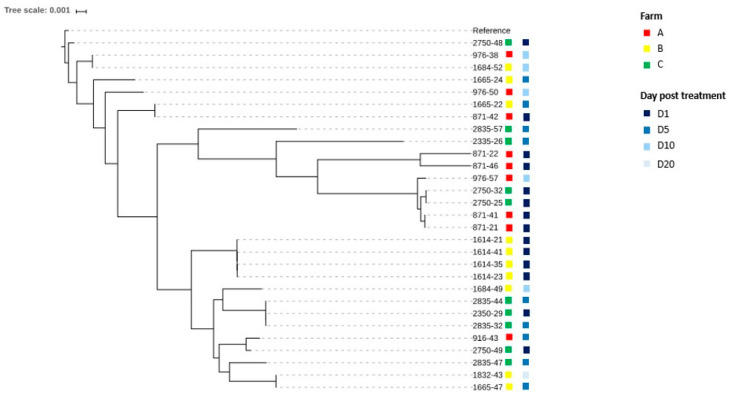
Maximum likelihood tree on SNPs of newly sequenced genomes of *Escherichia coli* isolates.

**Figure 5 animals-13-02590-f005:**
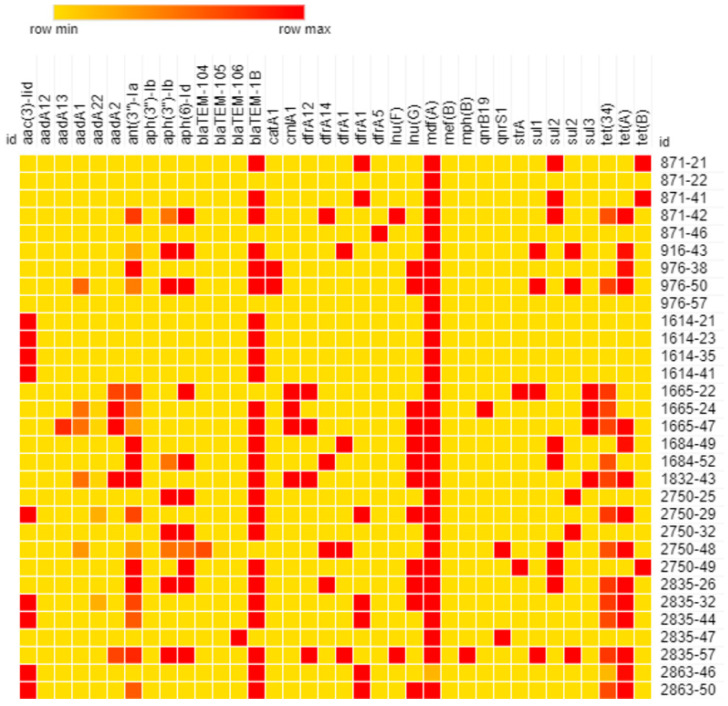
Heatmap of antimicrobial resistance genes in newly sequenced *E. coli* genomes (identity from 0% (yellow) to 100% (red)).

**Figure 6 animals-13-02590-f006:**
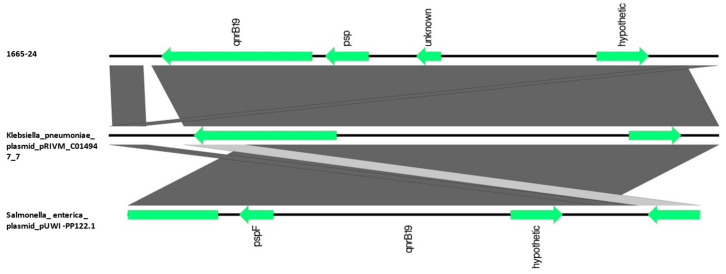
BLAST alignment of *qnrB19*-carrying contig in *E. coli* strain 1665-24 with plasmid pRIVM_C014947_7 (*Klebsiella pneumoniae* MT560070.1) and plasmid pUWI-PP122.1 (*Salmonella enterica* CP066326.1), performed and visualized with Easyfig 2.2.5 (https://mjsull.github.io/Easyfig/ (accessed on 28 July 2023)).

**Figure 7 animals-13-02590-f007:**
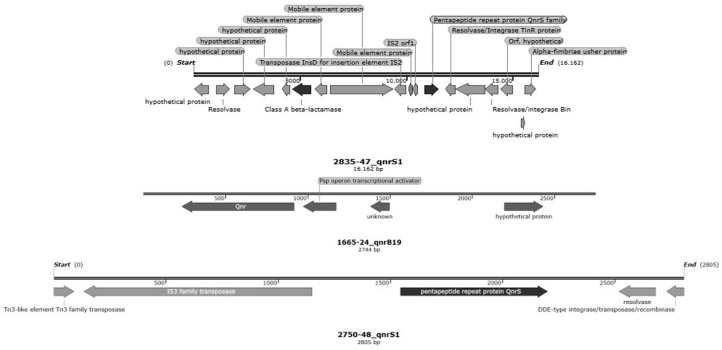
Quinolone-resistance-carrying contigs harboured by *E. coli* strains (*qnrS1*-carrying contigs of *E. coli* 2835-47 and 2750-48, and *qnrB19*-carrying contig of *E. coli* 1665-24).

**Table 1 animals-13-02590-t001:** Number of *E. coli* isolates collected from colisepticemia-affected broilers.

Day of Sampling	N° of *E. coli* Isolates				
	Lungs	Spleen	Heart	Cns	Total
D1	26	24	24	17	91
D5	23	8	8	2	41
D10	24	3	5	8	40
D24 *	5	0	1	1	7
Total	78	35	38	28	179

* exclusively from Farm B.

**Table 2 animals-13-02590-t002:** Serogroups of *E. coli* isolates collected from colisepticemia-affected broilers.

Day of Sampling	Serogroups								
	O2	O20	O78	O86	O128	O153	O157	Not Typable	Total
D1	1		36	5				49	91
D5						1	3	37	41
D10		1						39	40
D24			4		1			2	7
Total	1	1	40	5	1	1	3	127	179

**Table 3 animals-13-02590-t003:** MIC distribution (mg/L) of *E. coli* isolates against enrofloxacin (thick vertical lines representing the CLSI clinical breakpoints).

Day	MIC (mg/L)											
	0.016	0.03	0.06	0.12	0.25	0.5	1	2	4	8	16	32
D1	16 *	43	1	3	4	9	2	2	2	1	3	5 **
D5						1	1	6	2	12	12	7
D10	1 *			1	2	5		1	1	10	6	13
D24								1	1	4		1

* number of isolates showing MIC values less than the indicated dilution. ** number of isolates showing MIC values greater than the indicated dilution.

**Table 4 animals-13-02590-t004:** Fluoroquinolone resistance phenotypes and genotypes of newly sequenced *E. coli* genomes.

Farm	*E. coli* Genome ID	Day of Sampling	Phenotypes *	Genotypes	
				Genes	Mutations
A	871-21	1	R	-	*gyrA* S83L; *parC* D475E
	871-22	1	S	-	
	871-41	1	R	-	*gyrA* S83L
	871-42	1	R	-	*gyrA* S83L; *parC* S80R
	871-46	1	S	-	*gyrA* D87G
	916-43	5	R	-	*parC* S80I; *gyrA* S83L; *gyrA* D87N
	976-38	10	R	-	*parC* S80I; *gyrA* S83L
	976-50	10	R	-	*gyrA* S83L
	976-57	10	S	-	
B	1614-21	1	S	-	
	1614-23	1	S	-	
	1614-35	1	S	-	
	1614-41	1	S	-	
	1665-22	5	R	-	*parC* S80R; *gyrA* S83L
	1665-24	5	R	*qnrB19*	*parC* S80R; *gyrA* S83L
	1665-47	5	R	-	*parE* S458A; parC S80I; *gyrA* S83L; *gyrA* D87N
	1684-49	10	R	-	*gyrA* S83L; *gyrA* D87N; *parC* S80I
	1684-52	10	I	-	*gyrA* S83L; *parC* E84G
	1832-43	20	R	-	*parE* S458A; *parC* S80I; *gyrA* S83L; *gyrA* D87N
C	2750-29	1	R	-	*gyrA* S83L; *gyrA* D87N; *parC* S80I
	2750-25	1	S	-	
	2750-32	1	S	-	
	2750-48	1	R	*qnrS1*	*gyrA* S83L
	2750-49	1	S	-	*gyrA* S83L
	2835-26	5	R	-	*parC* S80I; *gyrA* S83L; *gyrA* D87Y
	2835-32	5	R	-	*gyrA* S83L; *gyrA* D87N; *parC* S80I
	2835-44	5	R	-	*parC* S80I; *gyrA* S83L; *gyrA* D87N
	2835-47	5	I	*qnrS1*	
	2835-57	5	R	-	*gyrA* S83L; *gyrA* D87N; *parC* S80I
	2863-46	10	R	-	*gyrA* S83L; *gyrA* D87N; *parC* S80I
	2863-50	10	R	-	*gyrA* S83L; *gyrA* D87N; *parC* S80I

* R, resistant; I, intermediate; S, susceptible.

**Table 5 animals-13-02590-t005:** Localization of antimicrobial-resistance-associated genes.

Farm	Isolate	Day of Sampling	Plasmid			Chromosome—AMR Genes
			Primary Cluster ID	Predicted Mobility	AMR Genes	
A	871-21	1	AF098	Non-mobilizable	*tet(B)*; *bla*TEM-1B	*mdf(A)*; *sul2*; *dfrA1*
	871-22	1	-	-	-	*mdf(A)*
	871-41	1	AA178	Conjugative	*bla*TEM-1B	*mdf(A)*; *sul2*; *tet(B)*; *dfrA1*
	871-42	1	AB595	Mobilizable	*aph(6)-Id*; *dfrA1*4; *aph(3”)-Ib*; *sul2*	*mdf(A)*; *Inu(F)*; *ant(3”)-Ia*
	871-42	1	AG600	Non-mobilizable	*tet(A)*	-
	871-46	1	AB233	Conjugative	*bla*TEM-1B	*mdf(A)*
	871-46	1	AA176	Conjugative	*dfrA5*	-
	916-43	5	AA176	Conjugative	*sul3*; *aph(3”)-Ib*; *aph(6)-Id*; *dfrA1*; *sul1*	*mdf(A)*; *tet(A)*; *bla*TEM-1B
	976-38	10	AB233	Non-mobilizable	*catA1*; *tet(A)*; *bla*TEM-1B	*mdf(A)*
			AD069	Non-mobilizable	*ant(3”)-Ia*; *lnu(G)*	
	976-50	10	AA619	Conjugative	*lnu(G)*; *bla*TEM-1B	*aadA1*
			AA374	Mobilizable	*tet(A)*; *sul2*; *aph(3”)-Ib*; *aph(6)-Id*; *catA1*	
	976-57	10	-	-	-	*mdf(A)*
B	1614-21	1	AA474	Conjugative	*aac(3)-IId*; *bla*TEM-1B	*mdf(A)*
	1614-23	1	AA175	Conjugative	*aac(3)-IId*; *bla*TEM-1B	*mdf(A)*
	1614-35	1	AA175	Conjugative	*aac(3)-IId*; *bla*TEM-1B	*mdf(A)*
	1614-41	1	AA474	Conjugative	*aac(3)-IId*; *bla*TEM-1B	*mdf(A)*
	1665-22	5	AA738	Conjugative	*strA*; *aph(6)-Id*; *sul3*; *sul1*; *cmlA1*	*mdf(A)*; *ant(3”)-Ia*; *aadA2*
	1665-22		novel	Non-mobilizable	*cmlA1*; *dfrA1*2	-
	1665-24	5	AG685	Non-mobilizable	*sul3*; *aadA2*; *cmlA1*	*mdf(A)*; *bla*TEM-1B; *lnu(G)*; *aadA1*
			AB042	Mobilizable	*qnrB19*	
	1665-47	5	AG685	Non-mobilizable	*sul3*; *dfrA1*2; *aadA2*; *cmlA1*	*mdf(A)*; *aadA1*3; *tet(A)*; *Inu(G)*; *bla*TEM-1B; *aasA1*
	1684-49	10	AA474	Conjugative	*sul2*; *tet(A)*; *bla*TEM-1B	*mdf(A)*; *lnu(G)*; *ant(3”)-Ia*; *dfrA1*
	1684-52	10	AA619	Conjugative	*bla*TEM-1B	*mdf(A)*; *tet(34)*
			AB595	Mobilizable	*aph(6)-Id*; *dfrA1*4; *aph(3”)-Ib*; *sul2*	
			AD068	Non-mobilizable	*ant(3”)-Ia_1*; *lnu(G)_1*	
	1832-43	24	AG658	Non-mobilizable	*sul3*; *cmlA1*; *aadA2*; *dfrA1*2	*mdf(A)*; *tet(34)*; *ant(3”)-Ia*; *tet(A)*; *lnu(G)*; *aadA1*
			AD448	Non-mobilizable	*bla*TEM-1B	
C	2750-25	1	AC120	Non-mobilizable	*aph(6)-Id*; *aph(3”)-Ib*; *sul2*; *tet(A)*; *bla*TEM-1B	*mdf(A)*
	2750-29	1	AA474	Conjugative	*tet(A)*; *dfrA1*; *aac(3)-IId*; *bla*TEM-1B	*mdf(A)*; *ant(3”)-Ia*
			novel	Conjugative	*Inu(G)*	
	2750-32	1	AA176	Conjugative	*aph(6)-Id*; *aph(3”)-Ib*; *sul2*; *bla*TEM-1B	*mdf(A)*; *tet(A)*
	2750-48	1	AB711	Conjugative	*bla*TEM-1B; *qnrS1*	*mdf(A)*; *tet(A)*
			AB595	Mobilizable	*aph(6)-Id*; *dfrA1*4; *aph(3”)-Ib*; *sul2*	
	2750-49	1	AA281	Conjugative	*bla*TEM-1B	*dfrA1_10*; *mdf(A)_1*; *sul2_2*
			AB193	Non-mobilizable	*aph(6)-Id*; *strA*; *ant(3”)-Ia*; *tet(B)*; *lnu(G)*	
	2835-26	5	AA179	Conjugative	*sul2*; *tet(A)*; *bla*TEM-1B	*aph(6)-Id*; *aph(3”)-Ib*; *lnu(G)*; *ant(3”)-Ia*; *dfrA1*4
	2835-32	5	AA474	Conjugative	*aac(3)-IId*; *tet(A)*; *bla*TEM-1B	*mdf(A)*; *dfrA1*; *ant(3”)-Ia*
			AA304	Conjugative	*lnu(G)*	
	2835-44	5	AA474	Conjugative	*tet(A)*; *dfrA1*; *aac(3)-IId*; *bla*TEM-1B	*mdf(A)*; *dfrA1*; *ant(3”)-Ia*
	2835-47	5	AB233	Conjugative	*bla*TEM-106; *qnrS1*	*mdf(A)*
	2835-57	5	AA738	Conjugative	*tet(A)*; *mph(B)*; *sul2*; *aph(3”)-Ib*; *aph(6)-Id*; *sul1*; *ant(3”)-Ia*; *dfrA1*; *dfrA1*2; *nu(F)*; *bla*TEM-1B	*mdf(A)*; *aadA2*
	2863-46	10	AA474	Conjugative	*tet(A)*; *aac(3)-IId*; *bla*TEM-1B	*dfrA1*; *mdf(A)*; *ant(3”)-Ia*
	2863-50	10	AA474	Conjugative	*aac(3)-IId*; *tet(A)*; *bla*TEM-1B; *lnu(G)*	*mdf(A)*; *dfrA1*; *ant(3”)-Ia*

## Data Availability

Read sequences are available at the European Nucleotide Archive under the study accession number PRJEB36793.

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
