# Peer review of "Genetic Diversity and Antimicrobial Resistance of Extraintestinal E. coli Populations Pre- and Post-Antimicrobial Therapy on Broilers Affected by Colisepticemia"

_animals, 2023, doi:10.3390/ani13162590_

Round 1

Reviewer 1 Report

1. In the study, the genetic diversity and antimicrobial resistance of E. coli isolated from spleen, percardium, lungs and brain during enrofloxacin therapy on broilers affected by colicepticemia was evaluated. The authors select three conventional farms, to observe the genetic diversity and antimicrobial resistance of E. coli before treatment and post-treatment. As the temporal variation of the E. coli population except the effect of treatment may exist. Thus, the genetic diversity and antimicrobial resistance of E. coli in each farm do not completely come from the effect of enrofloxacin therapy. This is my concern.    

2. In lines 93-95, why D1, D5, D10 and D24 on Farm B? while D1, D5, D10 on Farm A and Farm C.  

3. In lines 98-99, samples of lung, brain, pericardium and spleen were collected for E. coli isolation. While the organs are heart, spleen, lung and cns in Figure 1-3. Can heart represent pericardium? And what is cns? Can cns represent brain? The type of sample in not including liver, please explain the reason.

4. In line 114, the antisera of all 188 E. coli O antigens is purchased or self-made. please clarify.

5. In line 132, the intermediate and resistant of enrofloxacin should add unit.

6. In line 134, there were 179 E. coli isolates. And all isolates underwent molecular typing by PFGE. Why were the 31 isolates selected and whole genome sequenced? What was the selection standards ?

7. In line 183, Serotyping, 127 E. coli isolates were not typable among 179 E. coli isolates. In the discussion part, there was no explanation about why the large majority isolates was untypable. Maybe this is limited by the methodology.Do the authors try to use or explore other methods?

8. In line 198, af D1 in the same farm may be at D1 in the same farm.

9. Figures 1-3 are poor and low quality. Please, the authors must improve them, especially the percentage of similarity. Please clarify how to calulate the similarity.

10. In Table 3. MIC distribution (mg/l) of E. coli isolates against enrofloxacin (thick vertical lines representing the CLSI clinical breakpoints). where is the thick vertical lines? In the Materials and Methods part, the authors wrote  Twelve serial dilutions were tested from 0.016 to 32 mg/L and following clinical break points were used: susceptible S ≤ 0.25 mg/L, intermediate (I) if included in the range 0.5-1, resistant R ≥ 2. , thus the maximum dilution was 32 mg/L. however, in Table 3, there was 64 mg/L dilution. Besides, the Table was not conform to the regulation.

11. In line 276, what were the possible mechanisms of higher genetic diversity? Please discuss in the part of discussion.

12. In Figure 6. qnrS1 carrying contig of E. coli 2835-47. what about qnrS1 carrying contig of E. coli 2750-48? Does qnrB19 carry contig of E. coli 1665-24? If there is, please add a similar figure just like Figure 6.

13. In lines 327- 333, in my mind, this paragraph belongs to the part of discussion.

14. In my mind, please add a figure related to qnrB19 carrying contig of E. coli 1665-24, blast with plasmid pRIVM_C014947_7 of Klebsiella pneumoniae (accession number MT560070.1) and plasmid pUWI-PP122.1 of Salmonella enterica (accession number CP066326.1).

Author Response

Please see the attachment with detailed responses to all reviewer's comments. In the revised manuscript related modifications are highlighted in yellow.

Reviewer 2 Report

Dear authors,

 Thank you for submitting your well-designed and adequately documented research paper to Animals Journal. Before we can approve the manuscript for publication, please take into account the following comments and questions:

 The title of your article, "Genetic diversity...during antimicrobial therapy," may need revision as most of the E. coli isolates were collected before the administration of enrofloxacin to broilers. Please consider modifying the title accordingly.

 In line 19, you mention that the broilers were treated with ciprofloxacin or enrofloxacin. Please clarify this statement.

 On line 83, could you provide more information about the number of birds tested from each broiler farm? Additionally, do the broilers receive regular prophylactic antibiotic doses?

 Regarding line 108, could you explain how the 179 E. coli isolates were distributed among the three farms?

 You mentioned on line 176 that "one isolate was retained per organ." However, earlier you stated that "two colonies of E. coli were selected from each plate." Please clarify how you distinguished the isolates.

 The vertical representation of the CLSI breakpoints in Table 3 makes it difficult to read. Please consider reformatting the table for better clarity.

 Please address these comments and questions in your revised manuscript.

Thank you.

The language used in the manuscript could benefit from some minor improvements.

Author Response

Please see the attachment for detailed responses. In the revised manuscript related modifications are highlighted in green

Reviewer 3 Report

In their manuscript, the Authors describe the modifications observed in Escherichia coli strains isolated from poultry farms monitored during a colibacillosis outbreak, focusing especially on the effects of antibiotic treatments with enrofloxacin on their genomic diversity, before and after the treatment. The outcomes from this study evidenced a possible selection of resistant strains after treatments, with the detection of point mutations in some of the genes reported to be associated with fluoroquinolone resistance, as well as other genes.

The study was well designed and described, and the results  are clearly presented. Furthermore, the adopted approach allowed to collect comprehensive data, combining prevalence information by serological and molecular typing, as well as including more specific details through a genomic approach. Additionally, the topic of antimicrobial resistance in production animals is in line with the current direction of research about this field and it is definitely worth to be investigated. 

Therefore, I would suggest this manuscript to be accepted for publication. 

The only minor comment is about the need to double-check the use of italics for "E. coli" throughout the text.

Author Response

please see the attachement of detailed responses to reviewer's comments. 

Round 2

Reviewer 1 Report

11. In line 276, what were the possible mechanisms of higher genetic diversity? Please discuss in the part of discussion. Thank you for this comment that give us the possibility to better clarify this point. The higher genetic diversity might be associated to the disappearance of the pathogenic E. coli clones due to the antimicrobial treatment, and the emergence of different apathogenic E. coli ones. The disappearance of pathological lesions on selected organs after the treatment (data not shown) reinforce this hypothesis. A sentence was added at lines 283-286.   In my mind, the possible mechanisms of higher genetic diversity should be put in the part of discussion.   

Author Response

Dear Reviewer 1, thank you for your comment. We added a sentence on possible reasons of  higher genetic diversity in the part of discussion.